# Wealth Inequality as a Predictor of Subjective Health, Happiness and Life Satisfaction among Nepalese Women

**DOI:** 10.3390/ijerph15122836

**Published:** 2018-12-12

**Authors:** Zhifei He, Zhaohui Cheng, Ghose Bishwajit, Dongsheng Zou

**Affiliations:** 1School of Politics and Public Administration, Southwest University of Political Science & Law, Chongqing 401120, China; houis123@163.com; 2Chongqing Health Information Center, Chongqing 401120, China; czhbtx@163.com; 3School of International Development and Global Studies, University of Ottawa, Ottawa, ON K1N 6N5, Canada; brammaputram@gmail.com

**Keywords:** household wealth status, happiness, inequality, life satisfaction, self-reported health, subjective well-being

## Abstract

Socioeconomic status has shown to be associated with subjective health, well-being, satisfaction with overall life and estimation of happiness. The body of research concerning the question of whether higher economic status leads to better health and well-being are mostly from developed countries. The present study was therefore conducted among women in Nepal with an aim to investigate whether household wealth status is associated with satisfaction about (1) self-reported health, (2) happiness, and (3) life overall. *Methods:* Subjects were 5226 Nepalese women aged between 15 and 24 years. Cross-sectional data were extracted from round 5 of the Nepal Multiple Indicator Cluster Survey (NMICS), conducted in 2014, and analyzed using chi-square tests of association, bivariate and multivariable regression methods. *Results:* Wealth status was significantly associated with satisfaction about health, estimation of happiness and satisfaction. Compared with women in the poorest households, the odds of positive estimation about overall happiness were respectively 30% higher for poorer (*p* < 0.0001; 95% CI = 1.653–3.190), 80% higher for middle (*p* = 0.001; 95% CI = 1.294–2.522), 64% higher for richer (*p* = 0.006; 95% CI = 1.155–2.326), and 40% higher for richest households. The odds of reporting satisfaction about life were respectively 97% higher for poorer (*p* < 0.0001; 95% CI = 1.680–2.317), 41% higher for middle (*p* < 0.0001; 95% CI = 1.165–1.715), 62% higher for richer (*p* < 0.0001; 95% CI = 1.313–2.003), and 31% higher for richest households (*p* = 0.043; 95% CI = 1.008–1.700). *Conclusion:* Our results conclude that women in households with lower wealth status report poorer subjective health, quality of life and happiness. However, the findings need to be interpreted in light of the existing sociocultural conditions mediating the role of household wealth status on women’s lives.

## 1. Introduction

The body of research concerning the impact of wealth status on health and life outcomes have been rising steadily since last few decades. As more and more countries in the developing world are experiencing a steady rise in national GDP, higher rates of school enrolment, lower maternal and child mortality, there has been a renewed debate on whether or not economic development results in improvements in population health and well-being. However, a growing body of evidence suggests a significant correlation between individual wealth and health outcomes in both developed and developing societies. In general, people with higher socioeconomic status (SES) enjoy longer life spans and reduced risks of morbidity and mortality as wealth status improves [1,2]. With the growing understanding of the determinants of health, researchers and care-providers are placing increasing emphasis on financial well-being due to its role on overall psychosocial well-being of individuals. The importance of economic well-being on health outcomes, especially among the vulnerable population, is understood in terms of the necessary conditions required for maintaining a healthy lifestyle, avoiding the risk factors, and being able to seek proper medical care when facing health issues.

Several theories exist on the underlying mechanisms of the impact of SES on health outcomes. Lack of adequate wealth is likely to be associated with poor physical and mental health, livelihood and environmental standards with increased exposure to health deteriorating conditions. In the context of Nepal, socioeconomic conditions such as education, household financial situation, parental factors have been shown to play more prominent roles than risk factors in health and quality of life outcomes [1,2,3]. There is not enough evidence on health outcomes of women’s health and quality outcomes, however, low SES was reported to be associated with higher prevalence of malnutrition and infectious diseases among children, low utilization of maternal healthcare among women, and higher consumption of tobacco and alcohol among men [1,4,5]. Economic stress has also shown to be associated with chronic mental stress and depression, which in turn set in motion a chain of physiological events leading to adverse physiological conditions [5,6]. Some others proposed a cyclical relationship between health and wealth status, as lack of wealth can cause poorer health which in return causes a decline in wealth through low productivity and increased medical expenditures [7]. Intuitively, individual income is a good predictor of living standards which itself can impact overall health, happiness, and well-being among individuals. Moreover, wealth status is also associated with social position and networks which were found to be important determinants of health.

Apart from its association with health outcomes, wealth status in relation to the level of individual happiness and life satisfaction have also been very popular topics in behavioral health and psychological research. The branch of positive psychology is dedicated to studies on how to make living more satisfying rather than merely addressing the disease condition [8]. Subjective satisfaction has certain implications for health, as people with higher life satisfaction show better self-efficacy and adherence to healthy behavior [9]. Though socioeconomic wellbeing is regarded as a key indicator of quality of living, some argue that it may fail to give an appropriate measure of QoL because living standards are dependent on living and not in the possession of resources [10]. For instance, Bhutan has introduced a benchmark for national development based chiefly on happiness and well-being, that embraces the dimensions not only of material need, but also of people’s spiritual and social needs [11]. One’s perceived well-being is closely intertwined with psychosocial health, which in turn serve as a necessary precondition for financial well-being and QoL outcomes [12]. Individual life satisfaction, a closely related construct, though conceptually and methodologically different, shows an equally ambiguous correlation with wealth status. Happiness is a concept reflecting general satisfaction with one’s life and with oneself, and relates to the fulfilment of psychological needs (satisfaction with appearance, autonomy, respect, self-esteem, emotional well-being) which may not essentially be attainable through pecuniary means.

One striking difference between healthcare systems in developed and developing countries lies in the degree of importance they attach on the role of non-biological factors (e.g., socio-economic, cultural) on health and illness outcomes. The cultural environment of a society can influence health through various proximate and distal pathways e.g., health and dietary behavior, distribution of risk factors, equality of opportunities. Despite a large volume of theoretical and empirical studies showing the pathways through which wealth status and inequality affect health, happiness and well-being, there exists a huge research gap between developed and developing countries in this respect. The literature research revealed that almost all the studies on this topic are conducted on Europe and North American populations which indicates a potential oversight of the significance of the issue in developing countries. Therefore, in this study we aimed to focus on Nepal, one of the least developed of all Asian countries, to investigate how wealth inequality correlates with satisfaction about health, estimation of happiness and life overall. Till now, there was no nationwide survey on population well-being and happiness in Nepal. To overcome this lack, we extracted data from the most recent Nepal Multiple Indicator Cluster Survey (NMICS 2014) which included a special section on subjective well-being among women aged between 15 and 24 years. As a poverty and internal conflict-ridden country, Nepal has shown appreciable progress in terms of GDP growth, reduction in poverty rate, primary school enrolment rate along with several other health and macroeconomic indicators. However, the fledgling economy is already witnessing a rising wealth inequality with important implications for socioeconomic and political stability [13]. The results of the present study are expected to raise concerns regarding the wealth inequality in the context of health and societal well-being in the country.

## 2. Methods

### 2.1. About the Survey Program and Data Collection

Data for this study were collected from the fifth round of the Nepal Multiple Indicator Cluster Survey (NMICS). The survey was conducted from January to June 2014, and was implemented by the joint collaboration of Central Bureau of Statistics, National Planning Commission depends on the technical and financial support by United Nations Children’s Fund (UNICEF) Nepal. The survey aimed to provide information on women and children on various sociodemographic and health indicators to help monitor progress toward national and Millennium Development Goals (MDGs) and contribute to evidence-based sound policy making in the country. The survey was conducted in both urban and rural areas covering 15 sub-regions, and the sample was stratified by region in urban and rural areas. In total 13,000 households were selected for the survey and 12,405 were successfully interviewed producing a response rate of 98.5%. Four sets of questionnaires were used in the survey: (1) For household members, (2) for women aged between 15 and 49 years, (3) for children under five years of age, and (4) for water quality testing. Data for the present study was drawn from the women’s questionnaire. Further information on the sampling techniques and procedures for NMICS 2014 and data collection are available elsewhere [14].

### 2.2. Variables

The dependent variables included in this study were: (1) Satisfaction with health, (2) Estimation of happiness and (3) life satisfaction.

(1) Satisfaction with health: It was measured by the following question—‘How satisfied are you with your health?’ The possible answers were: (1) *Very satisfied*, (2) *Somewhat satisfied*, (3) *Neither satisfied nor unsatisfied*, (4) *Somewhat unsatisfied*, (5) *Very unsatisfied*.

For estimation of happiness and life satisfaction, answering the questions was assisted by displaying a card to the respondents with smiling faces corresponding to the degree of response categories: ‘*Very satisfied*’, ‘*somewhat satisfied*’, ‘*neither satisfied nor unsatisfied*’, ‘*somewhat* unsatisfied’ and ‘*very unsatisfied*’ [15].

(2) Estimation of overall happiness: Respondents were asked ‘Taking all things together, would you say you are (1) *very happy*, (2) *somewhat happy*, (3) *neither happy nor unhappy*, (4) *somewhat unhappy* (5) *very unhappy*?’

(3) Satisfaction with life overall: It was measured by the following question—‘How satisfied are you with your life, overall?’ with answers ranging from *very satisfied* to *very unsatisfied* the same way as the question on Satisfaction with health.

For this study, all three dependent variables were dichotomized in the following way: (1) Satisfied—*Very satisfied*, *somewhat satisfied*, and (2) Not satisfied—*Neither satisfied nor unsatisfied*, *somewhat unsatisfied*, *Very unsatisfied*.

Selection of explanatory variables was guided by a thorough literature review in PubMed. Based on the availability of the variables on the dataset, the following variables were included in this study: Age: *15–19/20–24 years*; Area: *Urban/Rural*; Ever attended school: *Yes/No*; Have children: *No/yes*; Listens to radio: *No/yes*; Watches TV: *No/yes*; Currently married: *Yes/No*; Ever smoked: *Yes/No*; Ever drank alcohol: *Yes/No*; Wealth index: *Poorest/Poorer/Middle/Richer/Richest* [15,16,17,18,19,20].

MICS surveys do not include any direct information on household income. However, the surveys provide a proxy measure of income-household wealth index which a composite indicator of household wealth status. Wealth index is constructed by appropriate statistical methods (principal components analysis) which is performed by assigning scores on individual household possessions, e.g., consumer goods, dwelling characteristics, water and sanitation to generate factor scores for each item. Households are then ranked based on individual scores to range between poorest, poorer, middle, richer and richest [14,21]. This method of assessing wealth status is applied universally in MICS surveys, and is used as an indicator of wealth gradient in population-based studies. For the present study, we used this variable as a proxy of individual wealth inequality.

### 2.3. Data Analysis

Datasets were checked for missing values and outliers. Baseline information about the study population were presented by descriptive statistics (Table 1). Chi-square tests of association were performed to check the association with the three dependent variables across all the explanatory variables (Table 2). Given the dichotomous nature of the outcome variables, a binary logistic regression technique was used to examine the independence of the association between satisfaction about health, estimation of happiness, and life satisfaction with the sociodemographic indicators. The results of the adjusted associations from multivariate analysis were presented as Odds Ratios and their 95% confidence intervals. An OR >1 indicates positive, <1 indicates negative, and 1 indicated no association. Three separate multivariate models were run for each of the outcome variables. Variables that showed association at 0.25 at the bivariate level were retained for multivariable analysis [22]. Variance inflation factor (VIF) was used as a measure of collinearity to ensure that none of the predictor variables in the final model were highly associated with each other. All statistical tests were two tailed and a *p*  <  0.05 was considered statistically significant (except for the cross-tabs). All analyses were performed with using SPSS® version 21.0 (IBM, Chicago, IL, USA).

## 3. Results

### 3.1. Descriptive Statistics

Baseline characteristics of the study population are presented in Table 1. The results show that the majority of the women were aged between 15 and 19 years, were of rural origin and had no children. The percentage of women that reported listening to radio and watching TV were respectively 64.6% and 60.5%. More than half of the women were currently married (54.9%). Prevalence of never smoking and drinking alcohol were 96.7% and 84.9%, respectively. Half of the women were reported to be living in poorest and poorer households and only 17% in richest households.

### 3.2. Chi-Square Tests of Associations

Results of cross tabulation in Table 2 shows that women who reported being satisfied with health and had positive estimation of happiness were more likely to be aged between 15 and 19 years, being of rural origin, attended school, and had children. Those who expressed satisfaction with life were also more likely to be of rural origin, attended school, had children, watched TV, and currently married. In the majority of the cases, women in the higher wealth groups were more likely to report being satisfied with health, overall happiness and satisfaction with life compared to women in lower wealth groups.

### 3.3. Association with Household Wealth Status and Satisfaction with Health, Estimation of Happiness, and Life Satisfaction

Results of both bivariate (Table 2) and multivariate regression analysis (Table 3, Table 4 and Table 5) showed that wealth index was strongly associated with satisfaction about health, estimation of happiness, and life satisfaction. Compared with the poorest households, living in poorer, middle, richer and richest households were associated with higher likelihoods of reporting better satisfaction with health, overall happiness, and life satisfaction.

Table 3 indicates that compared with the poorest households, women who lived in poorer households were 72% more likely (*p* < 0.0001; 95% CI = 1.351–2.200), middle wealth status 43% more likely (*p* = 0.002; 95% CI = 1.141–1.792), richer 44% more likely (*p* = 0.011; 95% CI = 1.090–1.919), and richest were 11% more likely (*p* = 0.346; 95% CI = 0.890–1.394) to be satisfied with health status. However, for richest households the association did not show any statistical significance.

Table 4 shows that compared with the women in the poorest households, the odds of positive estimation of about overall happiness were respectively 2.3 times (*p* < 0.0001; 95% CI = 1.653–3.190) higher among women in poorer, 80% higher among women in the middle (*p* = 0.001; 95% CI = 1.294–2.522), 64% higher among women in richer (*p* = 0.006; 95% CI = 1.155–2.326), and 40% (*p* = 0.045; 95% CI = 1.008–1.958) higher among women who lived in richest households.

According to the results shown in Table 5, compared with the women in the poorest households, the odds of reporting satisfaction about life were respectively 97% (*p* < 0.0001; 95% CI= 1.680–2.317), 41% (*p* < 0.0001; 95% CI =1.165–1.715), 62% (*p* < 0.0001; 95% CI =1.313–2.003), and 31% (*p* = 0.043; 95% CI =1.008–1.700) higher among women who lived in poorer, middle, richer, and richest households.

## 4. Discussion

With the pursuit of economic prosperity and increasingly unequal societies, the question of whether or not higher income influences happiness and well-being is a renewed concern across social, psychological, and behavioral sciences [22,23]. One of the most challenging tasks for well-being related research is finding an appropriate definition of the term and a universally applicable tool of measurement. Furthermore, interpretation of the research findings also needs special care depending on the variations in the conceptualization of and methodological approach to the issue. Subjective measurement of these terms has been found to resolve this issue to some extent and used extensively in social and psychological studies [24,25]. In the present study we attempted to investigate how subjective health, happiness, and life satisfaction are impacted by household wealth status among young women in Nepal.

Though there is a remarkable lack of research evidence from developing countries, our findings are in line with previous ones conducted in developed societies. Younger women of rural origin and having formal educational experience were more likely to be satisfied with health and estimation of happiness. The population of Nepal is still predominantly rural; however, the rate of urbanization is rapid, as seen in the neighboring countries in South Asia. Urbanization has been shown to be associated with higher socioeconomic inequality, more stressful life events, nuclear family, less social cohesion, all of which are associated with increased prevalence of stress and depression [15,26]. Women who were currently married and had no children were also more likely to report higher satisfaction about life and estimation of happiness compared to unmarried counterparts. Studies have shown the protective effects of marital status (conditional upon the quality of the marriage) on happy living and psychological satisfaction as it is involved with higher emotional support and well-being [17]. Individuals experiencing formal marriages tend to express higher levels of life satisfaction than those in other forms of marriages [27]. Interestingly, married individuals also tend to overestimate their health status, and the power of self-rated health to predict mortality among married individuals was reported be higher compared to that of unmarried ones [28].

Consistent with previous studies, we found a significant negative impact of household wealth inequality on subjective health, estimation of happiness and life satisfaction among the participants. The influence of wealth and income inequality on physical and mental health outcomes in the general population is well-documented across countries. Lack of wealth is commonly associated with poor nutritional status, adoption of unhealthy lifestyles, and risky behaviors which are associated with poor health outcomes [18,29]. In the context of women’s health in the South Asian countries, some noteworthy impacts of poverty and wealth inequality on public health include poor sanitation and environmental hygiene, and financial autonomy to access healthcare services. The situation is exacerbated by the deep-rooted gender inequality in the social value system which has important bearings on health and overall well-being of women. Being subject to social and material disadvantages, women face an increased susceptibility to poor health conditions and higher frequency of depression [19,20]. The situation needs to be addressed by strong social and political commitment and implementation of gender-sensitive policies to protect the rights of women across various domains of social and political life.

Compared with health, the association between wealth status and estimation of happiness and life satisfaction are less straightforward, and research findings are also highly mixed. Whether or not wealth can buy happiness is a highly debatable subject, and unlikely to have any definitive answer because of its contingency upon a plethora of non-pecuniary issues, e.g., quality of relationships, feeling of safety, social capital and mutual trust. Apart from the material and financial factors, health and well-being among women are also strongly influenced by intangible social goods such as gender equality and human rights. In the traditional South Asian society, women’s poorer psychosocial health and lower socioeconomic position are generally attributed to the patriarchal social structure and lack of decision-making autonomy [30]. The adverse impacts of gender inequality are likely to be compounded by the caste system that has been shown to be negatively associated with socioeconomic empowerment and reproductive rights [31]. Although the current data are not suitable to affirm the contribution of these factors to health and QoL outcomes among Nepalese women, the relationships are certainly worthy of investigation in making evidence-based policies for promotion of women’s health and overall well-being.

In light of the aforementioned discussion, and insight gained from current literature, there seems to be a concrete connection between wealth inequality and adverse outcomes on various health, social and psychological well-being related indicators. Therefore, it is recommended that national public health programs aiming at improving health and well-being among women should focus on devising policies to reduce wealth inequality among women. Though women with higher wealth status had higher rates of positive estimation of happiness and life overall, it should also be noted that these are complex constructs and not the result of any random indicator but of an interplay of a composite set of factors (e.g., an individual’s concept of well-being, family experience, socio-cultural environment). In addition, the relationship between wealth and health and well-being is not necessarily a linear one. Though higher wealth status has certain protective effects on health and happiness, the effect of rise in wealth status may not remain true after a certain level. Certain studies exist which found no association between wealth and mortality, and several other econometric analyses with little support to claim that low levels of wealth were associated with poor health [32]. This fact is also supported by the Easterlin paradox, maintaining that national economic growth does not always translate into increased happiness in the population [33]. Therefore, more cross-cutting research is needed to explore the underlying pathways that links wealth status to individual health and well-being instead of focusing exclusively on monetary gains [34].

To our knowledge, this is the first study to investigate the association between household wealth inequality and subjective health, happiness, and life satisfaction among women in Nepal. One particular strength of this study is that the data is nationally representative and presents the health and psychosocial situation of women. This should have important implications for addressing wealth inequality among women in the country in the context of health and social well-being. However, there are some important limitations which need to be noted. As the data were secondary, authors had no influence over the selection and measurement of the variables. The cross-sectional nature of the data also precludes making any causal inference between the explanatory and response variables. Measures of the outcome variables were self-reported, which is subject to biasness and recall error. Importantly, there was no direct information of the variable of household income, and was proxied by the construct of wealth index which is used universally as an indicator of household wealth status by MICS and DHS (Demographic and Health Survey) surveys. The selection of the explanatory variables was based on insights from research conducted on populations mainly from developed countries. Additionally, the sample consisted only of women ageing up to 24 years, and hence may not reflect the overall situation in the country. Last but not least, the data are cross-sectional, and the associations cannot be interpreted as causal mechanisms. Future studies should focus on a broader range of indicators of subjective well-being and include women of a broader age range.

## 5. Conclusions

Our results conclude that lower household wealth status has a significantly negative association with subjective health, estimation of happiness and life satisfaction. Subjective well-being is a strong predictor of individuals’ perception about life and their skills and efficacy to lead a normal and healthy life. However, improving socioeconomic status alone may not bring well-being and must be accompanied by creating gender-friendly sociocultural conditions congenial to creative and respectful living standards for women. This findings for this study call for policy actions at a national level, to address wealth inequality among women in an effort to ameliorate health and living standards of women in the increasingly unequal society.

## Figures and Tables

**Table 1 ijerph-15-02836-t001:** Baseline characteristics of the study population, Nepal Multiple Indicator Cluster Survey (MICS) 2014.

Variable	Frequency	Percentage (%)
Age		
15–19	2775	53.1
20–24	2451	46.9
Area		
Urban	1223	23.4
Rural	4003	76.6
Ever attended school		
Yes	4622	88.4
No	604	11.6
Have children		
No	3637	69.6
Yes	1589	30.4
Listens to radio		
No	1850	35.4
Yes	3376	64.6
Watches TV		
No	2062	39.5
Yes	3164	60.5
Currently married		
No	2355	45.1
Yes	2871	54.9
Ever smoked		
Yes	175	3.3
No	5051	96.7
Ever drank alcohol		
Yes	787	15.1
No	4436	84.9
Wealth index		
Poorest	1532	29.3
Poorer	1080	20.7
Middle	804	15.4
Richer	920	17.6
Richest	890	17.0

**Table 2 ijerph-15-02836-t002:** Percentage of women who reported being satisfied with health, overall happiness, and life across the explanatory variables, Nepal MICS 2014.

Variables	Satisfaction with Health	Positive Estimation of Overall Happiness	Overall Life Satisfaction
	79.9%	82.1%	80.8%
Age			
15–19	54.4%	54.1%	53.6%
20–24	45.6%	45.9%	46.4%
*p*-value	<0.0001	0.002	0.096
**Area**			
Urban	24.5%	24.1%	24.4%
Rural	75.5%	75.9%	75.6%
*p*-value	<0.0001	0.004	<0.0001
Ever attended school			
Yes	90.2%	89.9%	89.7%
No	9.8	10.1	10.3
*p*-value	<0.0001	<0.0001	<0.0001
Have children			
No	71.2%	70.9%	70.7%
Yes	28.8%	29.1%	29.3%
*p*-value	<0.0001	<0.0001	<0.0001
Listens to radio			
No	35.4%	35.5%	35.1%
Yes	64.6%	64.5%	64.9%
*p*-value	0.490	0.446	0.214
Watches TV			
No	38.4%	39.0%	38.6%
Yes	61.6%	61.0%	61.4%
*p*-value	0.001	0.059	0.004
Currently married			
No	43.4%	43.8%	44.2%
Yes	56.6%	56.2%	55.8%
*p*-value	<0.0001	<0.0001	0.006
Ever smoked			
No	96.8%	97.2%	96.8%
Yes	3.2%	2.8%	3.2%
*p*-value	0.112	<0.0001	0.124
Ever drank alcohol			
No	84.9%	85.6%	84.8%
Yes	15.1%	14.3%	15.1%
*p*-value	0.874	0.004	0.792
Wealth index			
Poorest	19.2 %	21.2%	18.3%
Poorer	13.3%	18.0%	18.05
Middle	23.9%	18.4%	15.0%
Richer	20.5%	20.5%	20.9%
Richest	21.1%	27.9%	27.8%
*p*-value	<0.0001	<0.0001	<0.0001

**Table 3 ijerph-15-02836-t003:** Association between household wealth index and satisfaction with health status, Nepal MICS 2014.

Wealth Index	*p*-Value	COR	95% CI	*p*-Value	AOR	95% CI
Poorest	-	-	-	-	-	-
Poorer	<0.0001	1.807	1.489–2.193	<0.0001	1.724	1.351–2.200
Middle	<0.0001	1.503	1.260–1.792	0.002	1.430	1.141–1.792
Richer	<0.0001	1.581	1.234–2.026	0.011	1.446	1.090–1.919
Richest	0.137	1.181	0.948–1.470	0.346	1.114	0.890–1.394

N.B. COR = Crude Odds ratio, AOR = Adjusted Odds ratio. Model adjusted for age, area, ever attending school, having children, watching TV, being currently married, ever smoking.

**Table 4 ijerph-15-02836-t004:** Association between household wealth index and positive estimation of overall happiness in life, Nepal MICS 2014.

Wealth Index	*p*-Value	COR	95% CI	*p*-Value	AOR	95% CI
Poorest	-	-	-	-	-	-
Poorer	<0.0001	2.011	1.620–2.496	<0.0001	2.296	1.653–3.190
Middle	<0.0001	1.666	1.293–2.146	0.001	1.806	1.294–2.522
Richer	0.004	1.593	1.163–2.182	0.006	1.639	1.155–2.326
Richest	0.028	1.388	1.036–1.859	0.045	1.405	1.008–1.958

N.B. COR = Crude Odds ratio, AOR = Adjusted Odds ratio. Model adjusted for age, area, ever attending school, having children, being currently married, watching TV, ever smoking, and ever drinking alcohol.

**Table 5 ijerph-15-02836-t005:** Association between household wealth index and overall life satisfaction, Nepal MICS 2014.

Wealth Index	*p*-Value	COR	95% CI	*P*-Value	AOR	95% CI
Poorest	-	-	-	-	-	-
Poorer	<0.0001	2.002	1.644–2.438	<0.0001	1.973	1.680–2.317
Middle	0.001	1.451	1.173–1.793	<0.0001	1.413	1.165–1.715
Richer	<0.0001	1.741	1.403–2.162	<0.0001	1.621	1.313–2.003
Richest	0.016	1.377	1.061–1.789	0.043	1.309	1.008–1.700

N.B. COR = Crude Odds ratio, AOR = Adjusted Odds ratio. Model adjusted for area, ever attending school, having children, watching TV, being currently married, ever smoking.

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
