# Peer review of "Wealth Inequality as a Predictor of Subjective Health, Happiness and Life Satisfaction among Nepalese Women"

_ijerph, 2018, doi:10.3390/ijerph15122836_

Round 1
Reviewer 1 Report
Due Oct 27, 2018
Review request: International Journal of Environmental Research and Public Health
Manuscript ID: ijerph-379778
Title: Wealth inequality as a predictor of subjective health, happiness and life satisfaction among Nepalese women
Synopsis
The aim of this study was "to investigate how wealth inequality correlates with satisfaction about health, estimation of happiness and life overall (Line 96-8)". The methods of analysis of the 2014 Nepal Multiple Indicator Cluster Survey (MICS) included the selection of purposive explanatory variables (age, attended school, have children, listen to radio, watch TV, married, smoking, drinking and wealth index) for individuals between ages 15 to 24 in 15 ecological zones of urban and rural sampling areas. The variables were analyzed by simple and multiple regression with variance inflation factors to check collinearity (not reported). Tables 3, 4, and 5 demonstrate that the "wealth index was strongly associated with satisfaction about health, estimation of happiness, and life satisfaction (line 191-2)". These results were also indicated by the odds ratio by using the poorest respondents as the control. The Discussion elaborates on the complexities of "complex constructs" with "interplay of a composite set of exogenous and endogenous factors". The limitations of the study were the nature of the secondary analysis and "the selection of the explanatory variables was based on insights from research conducted on population mainly from developed countries (Line 307-10)."
Reviewer's conflict of interest: None
Broad comments
The research question and methods of analysis were clear, and the conclusion is sound and valid. However, prior to reading, I was curious about three pieces of information: 1) who is female among these authors, 2) who has a background in sociology and anthropology, and 3) how they address the Caste system and arranged marriage (some forced marriage issues) in Nepal.
1) From the first names, the authors seem to be composed of a female first author and three male coauthors. Although some male researchers have a good understanding of feminist perspectives, when women's issues are addressed, I expect more female presence.
2) The school or departmental affiliation does not specify sociology, anthropology and/or psychology. The coauthors may have previous training and familiarity with Nepalese culture, but a quantitative analysis of global data may oversimplify the "non-pecuniary issues" by researchers without socio-cultural understanding.
3) Although general statements around social issues are discussed in the Introduction and Discussion sections, specifically the Caste system, Dalits and forced arranged marriage of Nepalese was not discussed. The lack of mention of the Caste system is due to the nature of the Nepalese MICS, which does not stratify data based on the Nepalese Caste. This study must also assume that marriage occurs usually within the same Caste class so that the family wealth index may not be changed by the marriage. However these assumptions were not clearly stated and need to be tested. A quantitative model alone that draws a conclusion may be premature. For example, Puri (2011) reported sexual violence within marriages among young women in Nepal (PMID: 21223603). A sociology researcher reported the disappearance (and therefore existence) of the Caste system may be related to Westernization in Nepal (PMID:15864977). Because this manuscript claims to be the first of its kind to provide information about young women in a developing country, the perspectives of analysis could be broader and encompass the socio-cultural perspective of the country. This may be the most important limitation of this study design. Statements such as "Women who were currently married and had no children were also more likely to report higher satisfaction about life and estimation of happiness compared to unmarried counterparts." (Line 240 – 2) is a correct statement from quantitative aggregates; however, it has to be carefully interpreted as this conclusion may support maintaining the traditional role of young women in this society who are forced into arranged marriage for economic reasons.
Originality/Novelty: This is a good research question that needs the attention of researchers.
Significance: From a quantitative perspective, the interpretation of the analysis is sound; however, the inherent weakness of the research design was not clearly addressed, i.e. ecological fallacy.
Quality of Presentation: Tables were simple and easy to understand.
Scientific Soundness: A set of national data with regression analyses may not be sufficient to draw a strong conclusion.
Interest to the Readers: Health disparity of women in developing countries is an appropriate research topic for this journal.
Overall Merit: This can be a shorter article reporting results of Nepalese MICS data without lengthy introduction and discussion. Additional quantitative analyses may be helpful to make the introduction and discussion more relevant.
English Level: It may be easier for readers if the authors can break up lengthy sentences.
Specific comments
Comments and other specific recommendations are listed below:
1. Abstract. Line 18 – 19 contains an incomplete sentence. The line 24 – 31 contains a sentence, "Compared with women in the poorest households, the odds of positive estimation about overall happiness were respectively 30% higher for poorer, 80% higher for middle, 64% higher for richer, and 40% higher for richest households; and that of reporting satisfaction about life were respectively 97% higher for poorer, 41% higher for middle, 62% higher for richer, and 31% higher for richest households." This is a 62-word sentence. Usually a sentence contains 15 – 25 words; try to limit to 35 words. Line 24, "the poorest households" must be defined as the lowest of the quintet or the sample who responded as in the lowest wealth index. The "existing socio-cultural conditions" is specifically the women's role in the society, which is likely a confounded factor in the wealth index.
2. Introduction. Thorough introduction that is well presented. Some speculations or "intuitive understanding" from developed cultures may not fit with developing countries with traditional cultural values. The Gross National Happiness Index, instead of GDP, was proposed by a neighboring country (Bhutan). The happiness index of Nepal may be worth mentioning as a supplemental analysis of health.
3. Methods. Although all variables seemed to be analyzed together in multiple regression equations, controlling the marriage variable may improve the coefficient of determinants for the perceived health, happiness and satisfaction.
4. Results. Table 2, the wealth index showed a bimodal distribution in happiness and satisfaction, i.e. the sample in the lowest wealth index reported happiness and life satisfaction more than the middle strata. Authors reported this in the Discussion section as "less straightforward and research finding are also highly mixed" (line 268).
5. Discussion. Line 245 – 8 "Interestingly married individuals...that of unmarried ones[30].", a comma ',' is missing between "...health status and the power...". Statements such as "Though women with higher wealth status in the present study tend to have higher rates of positive estimation of happiness and life overall, it should also be noted that these are complex constructs and not the results of any random indicator but of an interplay of a composite set of exogenous and endogenous factor e.g. individual's concept of well-being , educational attainment, gender equality, family experience, socio-cultural environment." These 68 words sound eloquent but are simply stating that unknown factors are likely present in this quantitative study. As stated earlier, I recommend keeping the word limit shorter per statement. I wonder why the data on male samples in the same age Nepalese was not comparatively analyzed. Young Nepalese men in the lower socioeconomic group are likely single, and this may serve as one of the comparators of the effect of marriage in young women.
6. Conclusion. I recommend avoiding the use of "predictor of..." because this regression analysis is not a causative analyses. I also recommend in the conclusion to simply state that this is a data analysis of the set of national surveys (NMICS) because understanding young Nepalese women requires a combination of other methods of analysis reported by other researchers.
End of review.

Author Response
Reviewer 1:
Broad comments
The research question and methods of analysis were clear, and the conclusion is sound and valid. However, prior to reading, I was curious about three pieces of information: 1) who is female among these authors, 2) who has a background in sociology and anthropology, and 3) how they address the Caste system and arranged marriage (some forced marriage issues) in Nepal.
Response: Dear reviewer, thanks for this comment. We have a fair share of female authors in the team, and all have solid background in social determinants of health. Regarding the latest comment, caste system was not a focus of this study, our data were quantitative analysis and were extracted from population-based health survey that collects no variable on caste system/arranged marriage. We do agree that these are important components when it comes to assessing women’s health and well-being, however we also hope that you are aware of the lack of political concern and public data on like issues in Nepal as well as in other countries across the region.
1) From the first names, the authors seem to be composed of a female first author and three male coauthors. Although some male researchers have a good understanding of feminist perspectives, when women's issues are addressed, I expect more female presence.
Response: Dear reviewer, we understand your concern in this respect. We did contact our female associates in Nepal to discuss the concept of quality of life and subjective health among women. Their comments were certainly helpful in carrying out the research, but they were not added on the author list to comply with authorship criteria.
2) The school or departmental affiliation does not specify sociology, anthropology and/or psychology. The coauthors may have previous training and familiarity with Nepalese culture, but a quantitative analysis of global data may oversimplify the "non-pecuniary issues" by researchers without socio-cultural understanding.
Response: Dear reviewer, thanks for your comments. Schools these days are essentially interdisciplinary, or at least many researchers are. Health and quality of life are increasingly being understood as a cross-cutting subject that is attracting researchers from diverse fields.
We made sure that the data being used with a proper understanding of the Nepalese sociocultural structure and interpreted in a manner that can assist in health and social work, and encourage more research in this area. Each study has its own limitations and we are aware of our own, nonetheless we hope that the present work is an important contribution to the literature, and despite the limitations arising from the nature of the data, the study advances the notion of perceived health and happiness among women, which was the main underlying motif of the authors.
3) Although general statements around social issues are discussed in the Introduction and Discussion sections, specifically the Caste system, Dalits and forced arranged marriage of Nepalese was not discussed. The lack of mention of the Caste system is due to the nature of the Nepalese MICS, which does not stratify data based on the Nepalese Caste. This study must also assume that marriage occurs usually within the same Caste class so that the family wealth index may not be changed by the marriage. However these assumptions were not clearly stated and need to be tested. A quantitative model alone that draws a conclusion may be premature. For example, Puri (2011) reported sexual violence within marriages among young women in Nepal (PMID: 21223603). A sociology researcher reported the disappearance (and therefore existence) of the Caste system may be related to Westernization in Nepal (PMID:15864977). Because this manuscript claims to be the first of its kind to provide information about young women in a developing country, the perspectives of analysis could be broader and encompass the socio-cultural perspective of the country. This may be the most important limitation of this study design. Statements such as "Women who were currently married and had no children were also more likely to report higher satisfaction about life and estimation of happiness compared to unmarried counterparts." (Line 240 – 2) is a correct statement from quantitative aggregates; however, it has to be carefully interpreted as this conclusion may support maintaining the traditional role of young women in this society who are forced into arranged marriage for economic reasons.
Response: Dear reviewer, we completely agree that these are genuine concerns and appreciate your perspectives. We also hope that you appreciate the complexities in interpreting issues as qualitative as sexual violence and caste system in the absence of supporting data.
The political economy of caste and marital system in India and Nepal have been evolving fast, and although not a ‘thing in the past’, the whole dynamics have started to look a lot different from what it used to only a few decades ago. The philosophy and social perception of marriage are changing and in the same vein the role of caste system is diminishing. We agree that these narratives are important and without these the findings look premature, but the fact remains that we can do only as much as we can do. From previous experience, some reviewers are hyper-speculative about any interpretations not grounded by data which is why we choose to rely on qualitative interpretations in moderation.
MICS surveys have only recently started to collect data on subjective health and QoL apart from the traditional topics, so it’s understandable that they cannot ask certain socioculturally sensitive data (even though they are necessary to understand the subjective construct) without changing the protocol and getting approved. This certainly makes the task harder for authors because they have to make assumptions to patch for the lack of contextual variables, while making unbacked assumptions has its own set of limitations. For academic reviewers, it’s equally hard to judge studies that suffering from such deficiencies. Nonetheless, we decided to undertake the study with the hope that it’ll open the avenue for QoL studies which has so far remained an underappreciated topic for research on South Asian women.
Originality/Novelty: This is a good research question that needs the attention of researchers.
Significance: From a quantitative perspective, the interpretation of the analysis is sound; however, the inherent weakness of the research design was not clearly addressed, i.e. ecological fallacy.
Response: Ecological fallacy was now discussed in the limitations section.
Quality of Presentation: Tables were simple and easy to understand.
Scientific Soundness: A set of national data with regression analyses may not be sufficient to draw a strong conclusion.
Interest to the Readers: Health disparity of women in developing countries is an appropriate research topic for this journal.
Overall Merit: This can be a shorter article reporting results of Nepalese MICS data without lengthy introduction and discussion. Additional quantitative analyses may be helpful to make the introduction and discussion more relevant.
Response: We truncated both the introduction and discussion part as appropriate. Data were exploited to adequate details, but we are open to new suggestions. Thank you!
English Level: It may be easier for readers if the authors can break up lengthy sentences.
Specific comments
Comments and other specific recommendations are listed below:
1. Abstract. Line 18 – 19 contains an incomplete sentence. The line 24 – 31 contains a sentence, "Compared with women in the poorest households, the odds of positive estimation about overall happiness were respectively 30% higher for poorer, 80% higher for middle, 64% higher for richer, and 40% higher for richest households; and that of reporting satisfaction about life were respectively 97% higher for poorer, 41% higher for middle, 62% higher for richer, and 31% higher for richest households." This is a 62-word sentence. Usually a sentence contains 15 – 25 words; try to limit to 35 words. Line 24, "the poorest households" must be defined as the lowest of the quintet or the sample who responded as in the lowest wealth index. The "existing socio-cultural conditions" is specifically the women's role in the society, which is likely a confounded factor in the wealth index.
Response: Dear reviewer, the sentence somehow got shortened during formatting. It was now corrected: household wealth status is associated with satisfaction about- 1) self-reported health, 2) happiness, and 3) life overall. Thank you.
The sentence was partitioned by sub-categories for better readability. Thank you.
We checked for multicollinearity and found that only occupation was strongly correlated with wealth index, and was removed from analysis. Thank you so much!
2. Introduction. Thorough introduction that is well presented. Some speculations or "intuitive understanding" from developed cultures may not fit with developing countries with traditional cultural values. The Gross National Happiness Index, instead of GDP, was proposed by a neighboring country (Bhutan). The happiness index of Nepal may be worth mentioning as a supplemental analysis of health.
Response: Dear reviewer, thanks for this insight. We agree that referring to GHN is more appropriate than referring to GDP in developed countries, we updated the text with reference. Thank you so much!
3. Methods. Although all variables seemed to be analyzed together in multiple regression equations, controlling the marriage variable may improve the coefficient of determinants for the perceived health, happiness and satisfaction.
Response: All the regression models were adjusted for the demographic covariates, including ‘current marital status’. Thank you so much!
4. Results. Table 2, the wealth index showed a bimodal distribution in happiness and satisfaction, i.e. the sample in the lowest wealth index reported happiness and life satisfaction more than the middle strata. Authors reported this in the Discussion section as "less straightforward and research finding are also highly mixed" (line 268).
Response: Dear reviewer, thank you for your comments. Indeed, both the wealth index and educational status measured by MICS often not tally well with the health indicators. Some researchers prefer using perceived ‘income sufficiency’ as a better indicator of economic well-being, but MICS surveys still do not include that measure.
5. Discussion. Line 245 – 8 "Interestingly married individuals...that of unmarried ones[30].", a comma ',' is missing between "...health status and the power...". Statements such as "Though women with higher wealth status in the present study tend to have higher rates of positive estimation of happiness and life overall, it should also be noted that these are complex constructs and not the results of any random indicator but of an interplay of a composite set of exogenous and endogenous factor e.g. individual's concept of well-being , educational attainment, gender equality, family experience, socio-cultural environment." These 68 words sound eloquent but are simply stating that unknown factors are likely present in this quantitative study. As stated earlier, I recommend keeping the word limit shorter per statement. I wonder why the data on male samples in the same age Nepalese was not comparatively analyzed. Young Nepalese men in the lower socioeconomic group are likely single, and this may serve as one of the comparators of the effect of marriage in young women.
Response: Dear reviewer, thank you for your suggestions. There were several punctuation issues which are now addressed. Thank you so much!.
This sentence was shortened suggested.
Male samples are being studied separately. There are some contextual variables available uniquely for male participants, and therefore are not comparable with the present analysis.
6. Conclusion. I recommend avoiding the use of "predictor of..." because this regression analysis is not a causative analyses. I also recommend in the conclusion to simply state that this is a data analysis of the set of national surveys (NMICS) because understanding young Nepalese women requires a combination of other methods of analysis reported by other researchers.
Response: Thanks for the suggestion. We used ‘predictor’ in lieu of ‘determinant’ based on the understanding that predictor doesn’t imply causality: All causal relationships are predictive, but not all predictive relationships are causal.
Reviewer 2 Report
Wealth Inequality as a Predictor of Subjective Health, Happiness and Life Satisfaction among Nepalese Women
A brief summary
This article presents the findings of a secondary analysis of an existing dataset from the NMICS undertaken in 2014. The analysis examined the association between wealth and happiness and life satisfaction amongst Nepalese women aged between 15 and 24 years. The study found there was an association between indicators of wealth and happiness in this group but noted the influence of wider socio-cultural factors, in particular gender inequality.
Broad comments
An interesting article, and important particularly because of the dearth of data from developing countries like Nepal. The article could be improved with better use of paragraphs and general editing. The most important aspect of the article, that is the implications and impacts of the wider socio-cultural environment, need to be more clearly linked to the analysis and findings. At the moment they seem like two separate discussions within the text. Perhaps somewhere in the introduction the sociocultural context, especially gender inequality could be introduced. Also, there is a need to clarify what is meant by ‘wealth inequality’ especially when used to denote an association with wealth and indicators of happiness. Wealth inequality is different to low or high wealth.
Specific comments
Para beginning line 89 This paragraph is a really important point but the issue about the lack of data on non-biological factors needs to be explained more clearly.
Line 97 ‘wealth inequality’ needs to be defined.
Line 223 Economic prosperity seems to be the wrong term to use here, do you mean the pursuit of economic growth within this first sentence?
Line 233 do you mean lack of evidence from developing countries?
Line 248 By setting out the sociocultural context earlier could mean that the alcohol and smoking questions could be placed more clearly – ie there are increasing NCDs in Nepal with smoking and alcohol big risk factors. Women have lower consumption but are impacted by the related behaviours of men, and this is significant because of their own lack of power and control because of the status of women in Nepal. So, just an example how perhaps the discussion can be made more impactful and related to context.
Line 257-266 This ‘para’ is important and should be more prominent and linked through the article.
Author Response
Reviewer 2:
A brief summary
This article presents the findings of a secondary analysis of an existing dataset from the NMICS undertaken in 2014. The analysis examined the association between wealth and happiness and life satisfaction amongst Nepalese women aged between 15 and 24 years. The study found there was an association between indicators of wealth and happiness in this group but noted the influence of wider socio-cultural factors, in particular gender inequality.
Response:
Broad comments
An interesting article, and important particularly because of the dearth of data from developing countries like Nepal. The article could be improved with better use of paragraphs and general editing. The most important aspect of the article, that is the implications and impacts of the wider socio-cultural environment, need to be more clearly linked to the analysis and findings. At the moment they seem like two separate discussions within the text. Perhaps somewhere in the introduction the sociocultural context, especially gender inequality could be introduced. Also, there is a need to clarify what is meant by ‘wealth inequality’ especially when used to denote an association with wealth and indicators of happiness. Wealth inequality is different to low or high wealth.
Response: Thanks indeed for the interesting insights.
We discussed the aspects of sociocultural environment and gender inequality as suggested.
Specific comments
Para beginning line 89 This paragraph is a really important point but the issue about the lack of data on non-biological factors needs to be explained more clearly.
Response: Dear reviewer, in this sentence we mentioned the role of non-biological factors. This comment was extended further to clarify the meaning. Thanks for the suggestion.
Line 97 ‘wealth inequality’ needs to be defined.
Response: We understand the point you raised here. Wealth inequality is a multidimensional issue and cannot be captured in terms household wealth status. We use this term for the lack of a better alternative, as it is widely used in studies based on DHS and MICS data. We have now added a new comment in the methods section to clarify this.
Line 223 Economic prosperity seems to be the wrong term to use here, do you mean the pursuit of economic growth within this first sentence?
Response: This was rephrased as suggested. Thank you so much!
Line 233 do you mean lack of evidence from developing countries?
Response: Yes, this was rephrased as well. Thank you so much!
Line 248 By setting out the sociocultural context earlier could mean that the alcohol and smoking questions could be placed more clearly – ie there are increasing NCDs in Nepal with smoking and alcohol big risk factors. Women have lower consumption but are impacted by the related behaviours of men, and this is significant because of their own lack of power and control because of the status of women in Nepal. So, just an example how perhaps the discussion can be made more impactful and related to context.
Response: Thanks for this comment. If we understand correctly, you are referring to the effects of passive smoking among women, on the rising prevalence of NCDs. We are not certain if the data are grained enough to make this case here, or by relating to the sociocultural norm that makes women vulnerable to passive smoking. Perhaps more so because data actually do not show any association between smoking with the outcome variables.
Line 257-266 This ‘para’ is important and should be more prominent and linked through the article.
Response: This paragraph was revised to better align with the context of the study. Thank you so much!
Round 2
Reviewer 1 Report
A prompt revision response was appreciated. The revision showed much improvement. I have no further recommendations except for some punctuation and typos in the revision.
Author Response
Response: Thanks indeed for your comments. We have edited the whole text to make sure the residual errors are addressed. Please let us know if you have any specific errors to report. Thank you.
Reviewer 2 Report
In this revision I would have expected to see more changes to the document than have been made. In particular more description is still needed of the sociocultural and health outcome context in Nepal, and the definition of relevant terms, within the introduction/background section. Significantly it is still unclear what wealth inequality refers to and how it might relate to the household level. It was also disappointing to see that little effort was made to make the text clearer through editing. As written currently, the paragraphs are very long, difficult to read and it is difficult to understand the main points being made. Also some of the changes that are made appear not to be integrated properly within the existing text.
Author Response
Response: Thanks indeed for your comments. We did our best to address the comments in the last revision. But it is important to know whether you find the responses adequate. So thanks for letting us know your specific concerns. This time we are revising better to make sure all the comments are adequate met.
Currently there is not enough evidence regarding the social determinants of health among Nepalese women, which prevents us from giving a solid account on the risk factors and health outcomes. We have now added some evidence in the introduction on maternal healthcare, health behaviour and child’s health determinants in Nepalese context to clarify the background better. Although these evidences are not very relevant to our study, they still help to understand how financial capacity can affect these particular determinants of health.
The paragraphs were edited to make shorter, more precise, and rephrase ambiguous terms to improve readability. Although its hard to say that a text can be free of errors, we tried make sure that the language is fluid and doesn’t compromise the readability of the article. Thanks again for your valuable time to read our paper.